# ON RELATING 'WHY?' AND 'WHY NOT?' EXPLANATIONS

## ABSTRACT

Explanations of Machine Learning (ML) models often address a 'Why?' question. Such explanations can be related with selecting feature-value pairs which are sufficient for the prediction. Recent work has investigated explanations that address a 'Why Not?' question, i.e. finding a change of feature values that guarantee a change of prediction. Given their goals, these two forms of explaining predictions of ML models appear to be mostly unrelated. However, this paper demonstrates otherwise, and establishes a rigorous formal relationship between 'Why?' and 'Why Not?' explanations. Concretely, the paper proves that, for any given instance, 'Why?' explanations are minimal hitting sets of 'Why Not?' explanations and vice-versa. Furthermore, the paper devises novel algorithms for extracting and enumerating both forms of explanations.

## 1 INTRODUCTION

The importance of devising mechanisms for computing explanations of Machine Learning (ML) models cannot be overstated, as illustrated by the fast-growing body of work in this area. A glimpse of the importance of explainable AI (XAI) is offered by a growing number of recent surveys and overviews Hoffman & Klein (2017); Hoffman et al. (2017); Biran & Cotton (2017); Montavon et al. (2018); Klein (2018); Hoffman et al. (2018a); Adadi & Berrada (2018); Alonso et al. (2018); Dosilovic et al. (2018); Hoffman et al. (2018b); Guidotti et al. (2019); Samek et al. (2019); Samek & Müller (2019); Miller (2019b;a); Anjomshoae et al. (2019); Mittelstadt et al. (2019); Xu et al. (2019).

Past work on computing explanations has mostly addressed *local* (or instance-dependent) explanations Ribeiro et al. (2016); Lundberg & Lee (2017); Ribeiro et al. (2018); Shih et al. (2018; 2019); Ignatiev et al. (2019a); Darwiche & Hirth (2020); Darwiche (2020). Exceptions include for example approaches that distill ML models, e.g. the case of NNs Frosst & Hinton (2017) among many others Ribeiro et al. (2016), or recent work on relating explanations with adversarial examples Ignatiev et al. (2019b), both of which can be seen as seeking *global* (or instance-independent) explanations. Prior research has also mostly considered model-agnostic explanations Ribeiro et al. (2016); Lundberg & Lee (2017); Ribeiro et al. (2018). Recent work on model-based explanations, e.g. Shih et al. (2018); Ignatiev et al. (2019a), refers to local (or global) model-agnostic explanations as *heuristic*, given that these approaches offer no *formal* guarantees with respect to the underlying ML model[1]. Examples of heuristic approaches include Ribeiro et al. (2016); Lundberg & Lee (2017); Ribeiro et al. (2018), among many others[2]. In contrast, local (or global) model-based explanations are referred to as *rigorous*, since these offer the strongest formal guarantees with respect to the underlying ML model. Concrete examples of such rigorous approaches include Shih et al. (2018); Tran & d'Avila Garcez (2018); Shih et al. (2019); Ignatiev et al. (2019a;b); Darwiche & Hirth (2020); Jha et al. (2019).

Most work on computing explanations aims to answer a 'Why prediction $\pi$?' question. Some work proposes approximating the ML model's behavior with a linear model Ribeiro et al. (2016); Lundberg & Lee (2017). Most other work seeks to find a (often minimal) set of feature value pairs which is sufficient for the prediction, i.e. as long as those features take the specified values, the prediction does not change. For rigorous approaches, the answer to a 'Why prediction $\pi$?' question has been referred to as PI-explanations Shih et al. (2018; 2019), abductive explanations Ignatiev et al. (2019a), but also as (minimal) sufficient reasons Darwiche & Hirth (2020); Darwiche (2020). (Hereinafter, we use the term *abductive explanation* because of the other forms of explanations studied in the paper.)

---

[1] A taxonomy of ML model explanations used in this paper is included in Appendix A.

[2] There is also a recent XAI service offered by Google: `https://cloud.google.com/explainable-ai/`, inspired on similar ideas Google (2019).

Another dimension of explanations, studied in recent work Miller (2019b), is the difference between explanations for 'Why prediction $\pi$?' questions, e.g., 'Why did I get the loan?', and for 'Why prediction $\pi$ and not $\delta$?' questions, e.g., 'Why didn't I get the loan?'. Explanations for 'Why Not?' questions, labelled by Miller (2019b) *contrastive* explanations, isolate a pragmatic component of explanations that *abductive explanations* lack. Concretely, an abductive explanation identifies a set of feature values which are sufficient for the model to make a prediction $\pi$ and thus provides an answer to the question 'Why $\pi$?' A constrastive explanation sets up a counterfactual link between what was a (possibly) *desired* outcome of a certain set of features and what was the observed outcome Bromberger (1962); Achinstein (1980). Thus, a contrastive explanation answers a 'Why $\pi$ and not $\delta$?' question Miller (2018); Dhurandhar et al. (2018); Mittelstadt et al. (2019).

In this paper we focus on the relationship between *local* abductive and contrastive explanations[3]. One of our contributions is to show how recent approaches for computing rigorous abductive explanations Shih et al. (2018; 2019); Ignatiev et al. (2019a); Darwiche & Hirth (2020); Darwiche (2020) can also be exploited for computing contrastive explanations. To our knowledge, this is new. In addition, we demonstrate that rigorous (model-based) local abductive and contrastive explanations are related by a minimal hitting set relationship [4], which builds on the seminal work of Reiter in the 80s Reiter (1987). Crucially, this novel hitting set relationship reveals a wealth of algorithms for computing and for enumerating contrastive and abductive explanations. We emphasize that it allows designing the first algorithm to *enumerate* abductive explanations. Finally, we demonstrate feasibility of our approach experimentally. Furthermore, our experiments show that there is a strong correlation between contrastive explanations and explanations produced by the commonly used SHAP explainer.

## 2 PRELIMINARIES

**Explainability in Machine Learning.** The paper assumes an ML model $\mathbb{M}$, which is represented by a finite set of first-order logic (FOL) sentences $\mathcal{M}$. (When applicable, simpler alternative representations for $\mathcal{M}$ can be considered, e.g. (decidable) fragments of FOL, (mixed-)integer linear programming, constraint language(s), etc.)[5] A set of features $\mathcal{F} = \{f_1, \ldots, f_L\}$ is assumed. Each feature $f_i$ is categorical (or ordinal), with values taken from some set $D_i$. An *instance* is an assignment of values to features. The space of instances, also referred to as *feature* (or *instance*) *space*, is defined by $\mathbb{F} = D_1 \times D_2 \times \ldots \times D_L$. (For real-valued features, a suitable interval discretization can be considered.) A (feature) literal $\lambda_i$ is of the form $(f_i = v_i)$, with $v_i \in D_i$. In what follows, a literal will be viewed as an atom, i.e. it can take value *true* or *false*. As a result, an instance can be viewed as a set of $L$ literals, denoting the $L$ distinct features, i.e. an instance contains a single occurrence of a literal defined on any given feature. A set of literals is consistent if it contains at most one literal defined on each feature. A consistent set of literals can be interpreted as a conjunction or as a disjunction of literals; this will be clear from the context. When interpreted as a conjunction, the set of literals denotes a *cube* in instance space, where the unspecified features can take any possible value of their domain. When interpreted as a disjunction, the set of literals denotes a *clause* in instance space. As before, the unspecified features can take any possible value of their domain.

The remainder of the paper assumes a classification problem with a set of classes $\mathbb{K} = \{\kappa_1, \ldots, \kappa_M\}$. A prediction $\pi \in \mathbb{K}$ is associated with each instance $X \in \mathbb{F}$. Throughout this paper, an ML model $\mathbb{M}$ will be associated with some logical representation (or encoding), whose consistency depends on the (input) instance and (output) prediction. Thus, we define a predicate $\mathcal{M} \subseteq \mathbb{F} \times \mathbb{K}$, such that $\mathcal{M}(X, \pi)$ is true iff the input $X$ is consistent with prediction $\pi$ given the ML model $\mathbb{M}$[6]. We further simplify the notation by using $\mathcal{M}_\pi(X)$ to denote a predicate $\mathcal{M}(X, \pi)$ for a concrete prediction $\pi$.

Moreover, we will compute *prime implicants* of $\mathcal{M}_\pi$. These predicates defined on $\mathbb{F}$ and represented as consistent conjunctions (or alternatively as sets) of feature literals. Concretely, a consistent

---

[3]In contrast with recent work Ignatiev et al. (2019b), which studies the relationship between *global* model-based (abductive) explanations and adversarial examples.

[4]A local abductive (resp. contrastive) explanation is a minimal hitting set of the set of all local contrastive (resp. abductive) explanations.

[5]$\mathcal{M}$ is referred to as the (formal) model of the ML model $\mathbb{M}$. The use of FOL is not restrictive, with fragments of FOL being used in recent years for modeling ML models in different settings. These include NNs Ignatiev et al. (2019a) and Bayesian Network Classifiers Shih et al. (2019), among others.

[6]This alternative notation is used for simplicity and clarity with respect to earlier work Shih et al. (2018); Ignatiev et al. (2019a;b). Furthermore, defining $\mathcal{M}$ as a predicate allows for multiple predictions for the same point in feature space. Nevertheless, such cases are not considered in this paper.

conjunction of feature literals $\tau$ is an implicant of $\mathcal{M}_\pi$ if the following FOL statement is true:

$$\forall (X \in \mathbb{F}).\tau(X) \rightarrow \mathcal{M}(X, \pi) \tag{1}$$

The notation $\tau \vDash \mathcal{M}_\pi$ is used to denote that $\tau$ an implicant of $\mathcal{M}_\pi$. Similarly, a consistent set of feature literals $\nu$ is the negation of an implicate of $\mathcal{M}_\pi$ if the following FOL statement is true:

$$\forall (X \in \mathbb{F}).\nu(X) \rightarrow (\vee_{\rho \neq \pi} \mathcal{M}(X, \rho)) \tag{2}$$

$\mathcal{M}_\pi \vDash \neg\nu$, or alternatively $(\nu \vDash \neg\mathcal{M}_\pi) \equiv (\nu \vDash \vee_{\rho \neq \pi} \mathcal{M}_\rho)$. An implicant $\tau$ (resp. implicate $\nu$) is called *prime* if none of its proper subsets $\tau' \subsetneq \tau$ (resp. $\nu' \subsetneq \nu$) is an implicant (resp. implicate).

Abductive explanations represent prime implicants of the decision function associated with some predicted class $\pi$[7].

**Analysis of Inconsistent Formulas.** Throughout the paper, we will be interested in formulas $\mathcal{F}$ that are *inconsistent* (or *unsatisfiable*), i.e. $\mathcal{F} \vDash \bot$, represented as conjunctions of clauses. Some clauses in $\mathcal{F}$ can be *relaxed* (i.e. allowed not to be satisfied) to restore consistency, whereas others cannot. Thus, we assume that $\mathcal{F}$ is partitioned into two first-order subformulas $\mathcal{F} = \mathcal{B} \cup \mathcal{R}$, where $\mathcal{R}$ contains the *relaxable* clauses, and $\mathcal{B}$ contains the *non-relaxable* clauses. $\mathcal{B}$ can be viewed as (consistent) background knowledge, which must always be satisfied.

Given an inconsistent formula $\mathcal{F}$, represented as a set of first-order clauses, we identify the clauses that are responsible for unsatisfiability among those that can be relaxed, as defined next[8].

**Definition 1 (Minimal Unsatisfiable Subset (MUS))** *Let $\mathcal{F} = \mathcal{B} \cup \mathcal{R}$ denote an inconsistent set of clauses ($\mathcal{F} \vDash \bot$). $\mathcal{U} \subseteq \mathcal{R}$ is a Minimal Unsatisfiable Subset (MUS) iff $\mathcal{B} \cup \mathcal{U} \vDash \bot$ and $\forall_{\mathcal{U}' \subsetneq \mathcal{U}}, \mathcal{B} \cup \mathcal{U}' \nvDash \bot$.*

Informally, an MUS provides the minimal information that needs to be added to the background knowledge $\mathcal{B}$ to obtain an inconsistency; it explains the causes for this inconsistency. Alternatively, one might be interested in correcting the formula, removing some clauses in $\mathcal{R}$ to achieve consistency.

**Definition 2 (Minimal Correction Subset (MCS))** *Let $\mathcal{F} = \mathcal{B} \cup \mathcal{R}$ denote an inconsistent set of clauses ($\mathcal{F} \vDash \bot$). $\mathcal{T} \subseteq \mathcal{R}$ is a Minimal Correction Subset (MCS) iff $\mathcal{B} \cup \mathcal{R} \setminus \mathcal{T} \nvDash \bot$ and $\forall_{\mathcal{T}' \subsetneq \mathcal{T}}, \mathcal{B} \cup \mathcal{R} \setminus \mathcal{T}' \vDash \bot$.*

A fundamental result in reasoning about inconsistent clause sets is the minimal hitting set (MHS) duality relationship between MUSes and MCSes Reiter (1987); Birnbaum & Lozinskii (2003): *MCSes are MHSes of MUSes and vice-versa.* This result has been extensively used in the development of algorithms for MUSes and MCSes Bailey & Stuckey (2005); Liffiton & Sakallah (2008); Liffiton et al. (2016), and also applied in a number of different settings. Recent years have witnessed the proposal of a large number of novel algorithms for the extraction and enumeration of MUSes and MCSes Bacchus & Katsirelos (2015); Liffiton et al. (2016); Grégoire et al. (2018); Bendík et al. (2018). Although most work addresses propositional theories, these algorithms can easily be generalized to any other setting where entailment is monotonic, e.g. SMT de Moura & Bjørner (2008).

**Running Example.** The following example will be used to illustrate the main ideas.

**Example 1** *We consider a textbook example Poole & Mackworth (2010)[Figure 7.1, page 289] addressing the classification of a user's preferences regarding whether to read or to skip a given book. For this dataset, the set of features is:*

$$\{ \mathsf{A(uthor)}, \mathsf{T(hread)}, \mathsf{L(ength)}, \mathsf{W(hereRead)} \}$$

*All features take one of two values, respectively* $\{\mathsf{known, unknown}\}$, $\{\mathsf{new, followUp}\}$, $\{\mathsf{long, short}\}$, *and* $\{\mathsf{home, work}\}$. *An example instance is:* $\{(\mathsf{A = known}), (\mathsf{T = new}), (\mathsf{L = long}), (\mathsf{W = home})\}$ *This instance is identified as $e_1$ Poole & Mackworth (2010) with prediction* skips. *Figure 1a shows a possible decision tree for this example Poole & Mackworth (2010)[9]. The decision tree can be represented as a set of rules as shown in Figure 1b[10].*

---

[7]By definition of prime implicant, abductive explanations are sufficient reasons for the prediction. Hence the names used in recent work: abductive explanations Ignatiev et al. (2019a), PI-explanations Shih et al. (2018; 2019) and sufficient reasons Darwiche & Hirth (2020); Darwiche (2020).

[8]The definitions in this section are often presented for the propositional case, but the extension to the first-order case is straightforward.

[9]The choice of a decision tree aims only at keeping the example(s) presented in the paper as simple as possible. The ideas proposed in the paper apply to *any* ML model that can be represented with FOL. This encompasses *any* existing ML model, with minor adaptations in case the ML model keeps state.

[10]The abbreviations used relate with the names in the decision tree, and serve for saving space.

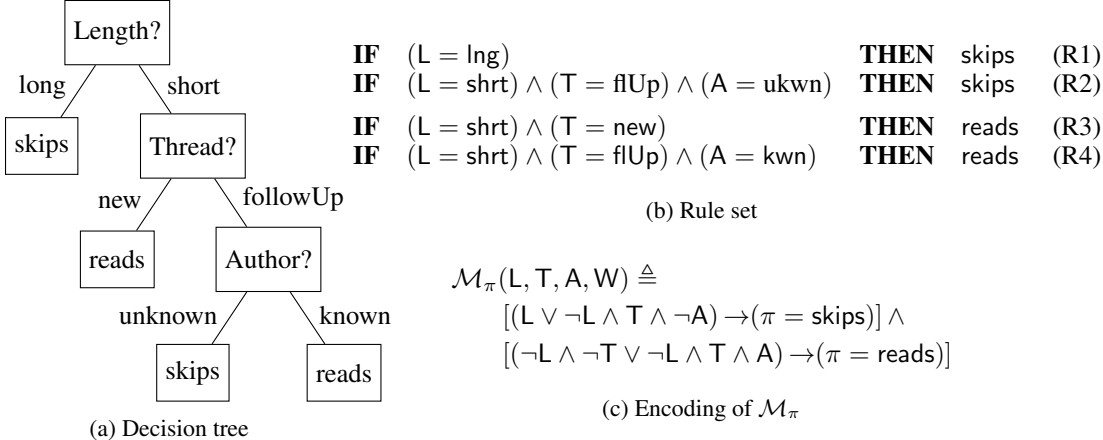

Figure 1: Running example Poole & Mackworth (2010)

Our goal is to reason about the ML model, i.e. to implement model-based reasoning, so we need to propose a logical representation for the ML model.

**Example 2** *For implementing model-based reasoning, we need to develop an encoding in some suitable fragment of FOL [11]. 0-place predicates [12] are used for* L, T, A *and* W, *as follows. We will associate* (L = long) *with* L *and* (L = short) *with* ¬L. *Similarly, we associate* (T = new) *with* ¬T, *and* (T = followUp) *with* T. *We associate* (A = known) *with* A *and* (A = unknown) *with* ¬A. *Furthermore, we associate* (W = home) *with* ¬W *and* (W = work) *with* W. *An example encoding is shown in Figure 1c. The explicit values of* π *are optional (i.e. propositional values could be used) and serve to illustrate how non-propositional valued could be modeled.*

## 3 CONTRASTIVE VS. ABDUCTIVE EXPLANATIONS

Recent work Shih et al. (2018; 2019); Ignatiev et al. (2019a); Darwiche (2020) proposed to relate model-based explanations with prime implicants. All these approaches compute a set of feature values which, if unchanged, are sufficient for the prediction. Thus, one can view such explanations as answering a 'Why?' question: *the prediction is the one given, as long as some selected set of feature values is the one given.* In this paper, such explanations will be referred to as *abductive explanations*, motivated by one of the approaches used for their computation Ignatiev et al. (2019a).

### 3.1 DEFINING ABDUCTIVE EXPLANATIONS (AXPS)

As indicated earlier in the paper, we focus on *local model-based* explanations.

**Definition 3 (Abductive Explanation)** *Given an instance* τ, *with a prediction* π, *and an ML model represented with a predicate* $\mathcal{M}_\pi$, *i.e.* $\tau \vDash \mathcal{M}_\pi$, *an* abductive explanation *is a minimal subset of literals of* τ, σ ⊆ τ, *such that* $\sigma \vDash \mathcal{M}_\pi$.

**Example 3** *With respect to Example 1, let us consider the instance* (A = known, T = new, L = short, W = work), *which we will represent instead as* (A, ¬T, ¬L, W), *corresponding to prediction* π = reads. *By inspection of the decision tree (seeFigure 1a), a possible answer to the 'Why pred.* reads*?' question is:* {¬L, ¬T}. *In this concrete case we can conclude that this is the only abductive explanation, again by inspection of the decision tree.*

### 3.2 DEFINING CONTRASTIVE EXPLANATIONS (CXPS)

As Miller (2019b) notes, contrastive explanations are,

> *"sought in response to particular counterfactual cases... That is, people do not ask why event P happened,but rather why event P happened instead of some event Q."*

---

[11]Depending on the ML problem, more expressive fragments of FOL logic could be considered Kroening & Strichman (2016). Well-known examples include real, integer and integer-real arithmetic, but also nonlinear arithmetic Kroening & Strichman (2016).

[12]Which in this case are used as propositional variables.

As a result, we are interested in providing an answer to the question 'Why $\pi$ and not $\delta$?', where $\pi$ is the prediction given some instance $\tau$, and $\delta$ is some other (desired) prediction.

**Example 4** *We consider again Example 1, but with the instance specified in Example 3. A possible answer to the question 'Why pred.* reads *and not pred.* skips*??' is* {L}*. Indeed, given the input instance* (A, ¬T, ¬L, W)*, if the value of feature* L *changes from* short *to* long*, and the value of the other features remains unchanged, then the prediction will change from* reads *to* skips*.*

The following definition of a (local model-based) contrastive explanation captures the intuitive notion of the contrastive explanation discussed in the example above.

**Definition 4 (Contrastive Explanation)** *Given an instance $\tau$, with a prediction $\pi$, and an ML model represented by a predicate $\mathcal{M}_\pi$, i.e. $\tau \vDash \mathcal{M}_\pi$, a* contrastive explanation *is a minimal subset of literals of $\tau$, $\rho \subseteq \tau$, such that $\tau \setminus \rho \nvDash \mathcal{M}_\pi$.*

This definition means that, there is an assignment to the features with literals in $\rho$, such that the prediction differs from $\pi$. Observe that a CXp is defined to answer the following (more specific) question 'Why (pred. $\pi$ and) not $\neg\pi$?'. The more general case of answering the question 'Why (pred. $\pi$ and) not $\delta$?' will be analyzed later.

### 3.3 Relating Abductive & Contrastive Explanations

The previous section proposed a rigorous, model-based, definition of contrastive explanation. Given this definition, one can think of developing dedicated algorithms that compute CXps using a decision procedure for the logic used for representing the ML model. Instead, we adopt a simpler approach. We build on a fundamental result from model-based diagnosis Reiter (1987) (and more generally for reasoning about inconsistency Birnbaum & Lozinskii (2003); Bailey & Stuckey (2005)) and demonstrate a similar relationship between AXps and CXps. In turn, this result reveals a variety of novel algorithms for computing CXps, but also offers ways for enumerating both CXps and AXps.

**Local Abductive Explanations (AXps).** Consider a set of feature values $\tau$, s.t. the predicion is $\pi$, for which the notation $\tau \vDash \mathcal{M}_\pi$ is used. We will use the equivalent statement, $\tau \wedge \neg \mathcal{M}_\pi \vDash \bot$. Thus,

$$\tau \wedge \neg \mathcal{M}_\pi \tag{3}$$

is inconsistent, with the background knowledge being $\mathcal{B} \triangleq \neg \mathcal{M}_\pi$ and the relaxable clauses being $\mathcal{R} \triangleq \tau$. As proposed in Shih et al. (2018); Ignatiev et al. (2019a), a (local abductive) explanation is a subset-minimal set $\sigma$ of the literals in $\tau$, such that, $\sigma \wedge \neg \mathcal{M}_\pi \vDash \bot$. Thus, $\sigma$ denotes a subset of the example's input features which, no matter the other feature values, ensure that the ML model predicts $\pi$. Thus, any MUS of equation 3 is a (local abductive) explanation for $\mathbb{M}$ to predict $\pi$ given $\tau$.

**Proposition 1** *Local model-based abductive explanations are MUSes of the pair $(\mathcal{B}, \mathcal{R})$, $\tau \wedge \neg \mathcal{M}_\pi$, where $\mathcal{R} \triangleq \tau$ and $\mathcal{B} \triangleq \neg \mathcal{M}_\pi$.*

**Example 5** *Consider the ML model from Example 1, the encoding from Example 2, and the instance* {A, ¬T, L, ¬W}*, with prediction $\pi =$ skips (wrt Figure 1, we replace* skips = skips *with* **true** *and* skips = reads *with* **false***). We can thus confirm that $\tau \vDash \mathcal{M}_\pi$. We observe that the following holds:*

$$A \wedge \neg T \wedge L \wedge \neg W \vDash [(L \vee \neg L \wedge T \wedge \neg A) \to \textbf{true}] \wedge [(\neg L \wedge \neg T \vee \neg L \wedge T \wedge A) \to \textbf{false}] \tag{4}$$

*which can be rewritten as,*

$$A \wedge \neg T \wedge L \wedge \neg W \wedge [(L \vee \neg L \wedge T \wedge \neg A) \wedge \neg \textbf{true}] \vee [(\neg L \wedge \neg T \vee \neg L \wedge T \wedge A) \wedge \neg \textbf{false}] \tag{5}$$

*It is easy to conclude that equation 5 is inconsistent. Moreover, $\sigma = $ (L) denotes an MUS of equation 5 and denotes one abductive explanation for why the prediction is* skips *for the instance $\tau$.*

**Local Contrastive Explanations (CXps).** Suppose we compute instead an MCS $\rho$ of equation 3, with $\rho \subseteq \tau$. As a result, $\bigwedge_{l \in \tau \setminus \rho}(l) \wedge \neg \mathcal{M}_\pi \nvDash \bot$ holds. Hence, assigning feature values to the inputs of the ML model is consistent with a prediction that is *not* $\pi$, i.e. a prediction of some value other than $\pi$. Observe that $\rho$ is a subset-minimal set of literals which causes $\tau \setminus \rho \wedge \neg \mathcal{M}_\pi$ to be satisfiable, with any satisfying assignment yielding a prediction that is not $\pi$.

**Proposition 2** *Local model-based contrastive explanations are MCSes of the pair $(\mathcal{B}, \mathcal{R})$, $\tau \wedge \neg \mathcal{M}_\pi$, where $\mathcal{R} \triangleq \tau$ and $\mathcal{B} \triangleq \neg \mathcal{M}_\pi$.*

**Example 6** *From equation 3 and equation 5 we can also compute $\rho \subseteq \tau$ such that $\tau \setminus \rho \wedge \neg \mathcal{M}_\pi \nvDash \bot$. For example $\rho = (\mathsf{L})$ is an MCS of equation 5* [13]. *Thus, from $\{\mathsf{A}, \neg \mathsf{T}, \neg \mathsf{W}\}$ we can get a prediction other than* skips, *by considering feature value* $\neg \mathsf{L}$.

**Duality Among Explanations.** Given the results above, and the hitting set duality between MUSes and MCSes Reiter (1987); Birnbaum & Lozinskii (2003), we have the following.

**Theorem 1** *AXps are MHSes of CXps and vice-versa.*

*Proof.* Immediate from Definition 3, Definition 4, Proposition 1, Proposition 2, and Theorem 4.4 and Corollary 4.5 of Reiter (1987). □

Proposition 1, Proposition 2, and Theorem 1 can now serve to exploit the vast body of work on the analysis of inconsistent formulas for computing both contrastive and abductive explanations and, arguably more importantly, to enumerate explanations. Existing algorithms for the extraction and enumeration of MUSes and MCSes require minor modications to be applied in the setting of AXps and CXps (The resulting algorithms are briefly summarized in Appendix B. Interestingly, a consequence of the duality is that computing an abductive explanation is *harder* than computing a contrastive explanation in terms of the number of calls to a decision procedure Appendix B.).

**Discussion.** As observed above, the contrastive explanations we are computing answer the question: 'Why ($\pi$ and) not $\neg\pi$?'. A more general contrastive explanation would be 'Why ($\pi$ and) not $\delta$, with $\pi \neq \delta$?' Miller (2019b). Note that, since the prediction $\pi$ is given, we are only interested in changing the prediction to either $\neg\pi$ or $\delta$. We refer to answering the first question as a *basic* contrastive explanation, whereas answering the second question will be referred to as a *targeted* contrastive explanation, and written as $\mathrm{CXp}_\delta$. The duality result between AXps and CXps in Theorem 1 applies *only* to basic contrastive explanations. Nevertheless, the algorithms for MCS extraction for computing a basic CXp can also be adapted to computing targeted CXps, as follows. We want a pick of feature values such that the prediction is $\delta$. We start by letting all features to take any value, and such that the resulting prediction is $\delta$. We then iteratively attempt to fix feature values to those in the given instance, while the prediction remains $\delta$. This way, the set of literals that change value are a subset-minimal set of feature-value pairs that is sufficient for predicting $\delta$. Finally, there are crucial differences between the duality result established in this section, which targets local explanations, and a recent result Ignatiev et al. (2019b), which targets *global* explanations. Earlier work established a relation between prime implicants and implicates as a way to relate global abductive explanations and so-called counterexamples. In contrast, we delved into the fundamentals of reasoning about inconsistency, concretely the duality between MCSes and MUSes, and established a relation between model-based *local* AXps and CXps.

## 4 EXPERIMENTAL EVALUATION

This section details the experimental evaluation to assess the practical feasibility and efficiency of the enumeration of abductive and contrastive explanations for a few real-world datasets, studied in the context of explainability and algorithmic fairness. To perform the evaluation, we adapt powerful algorithms for enumeration MCSes or MCSes and MUSes to find all abductive and contrastive explanations Bailey & Stuckey (2005); Liffiton & Sakallah (2008); Grégoire et al. (2018); Bendík et al. (2018) [14]. Algorithm 1 and Algorithm 2 in Appendix B show our adaptations of MCS (resp. MCS and MUS) enumeration algorithms to the enumeration of CXps (resp. AXps and CXps).

**Enumeration of CXps.** These experiments demonstrate a novel, unexpected practical use case of CXps enumeration algorithms. In particular, we show that our method gives a *new fine-grained view* on both global and local standard explanations extracted from ML models. The goal of these experiments is to *gain better understanding of existing explainers* rather than generate all CXps for a given input. We conduct two sets of experiments. The first experiment, called "real vs fake", distinguishes real from fake images. A dataset contains two classes of images: (a) original MNIST digits and (b) fake MNIST digits produced by a standard DCGAN model Radford et al. (2016) (see Figure 2a and Figure 2g for typical examples). The second experiment, called "3 vs 5 digits", uses a dataset that contains digits "3" and "5" from the standard MNIST dataset (discussed in Appendix C.1). Next, we discuss the results of the "real vs fake" experiment in details (Figure 2). For "real vs fake",

---

[13] Although in general not the case, in Example 5 and Example 6 an MUS of size 1 is also an MCS of size 1.

[14] The prototype and the experimental setup are available at `https://github.com/xdual/xdual`.

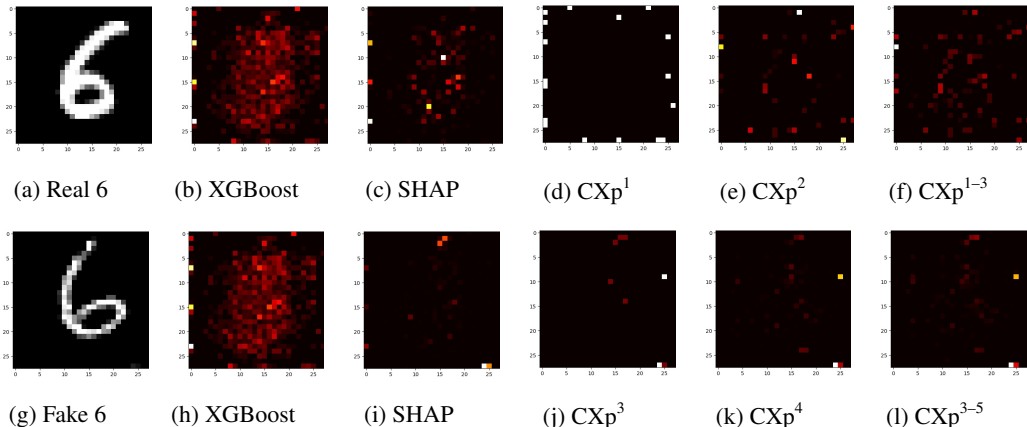

(a) Real 6    (b) XGBoost    (c) SHAP    (d) CXp$^1$    (e) CXp$^2$    (f) CXp$^{1-3}$

(g) Fake 6    (h) XGBoost    (i) SHAP    (j) CXp$^3$    (k) CXp$^4$    (l) CXp$^{3-5}$

Figure 2: The 'real vs fake' images. The first row shows results for the real image 6; the second – results for the fake image 6. The first column shows examples of inputs; the second – heatmaps of XGBoost's important features; the third – heatmaps of SHAP's explanation. Last three columns show heatmaps of CXp of different cardinality. The brighter pixels are more influential features.

we train an XGBoost model Chen & Guestrin (2016) with 100 trees of depth 6 (accuracy 0.85/0.80 on train/test sets). We quantized images so that each pixel takes a value between 0 and 15, image pixels are categorical features in the model.

*Brief overview of the SHAP explainer.* Given a classifier $f$ and an explainer model $g$, SHAP aims to train $g$ be similar to $f$ in the neighborhood of some given point $x$. The objective function for SHAP is designed so that: (1) $g$ approximates the behavior of the black box $f$ accurately within the vicinity of $x$, and (2) $g$ achieves lower complexity and is interpretable: $\xi(x) = \arg\min_{g \in G} L(\pi_x, g, f) + \Omega(g)$, where the loss function $L$ is defined to minimize the distance between $f$ and $g$ in the neighborhood of $x$ using a weight function $\pi_x$ and $\Omega(g)$ quantifies the complexity of $g$; $\Omega(g)$ and $\pi_x$ are defined based on game-theoretic notions (Lundberg & Lee, 2017).

*Global and local explainers.* We start by discussing our results on a few samples (Figure 2a and Figure 2g). First, we extract important features provided by XGBoost. As these features are *global* for the model, they are the same for all inputs (Figure 2b and Figure 2h are identical for real and fake images). Figure 2b shows that these important features are *no very informative* for this dataset as these pixels form a blob of pixels that cover an image. Then we compute an image-specific explanation using the standard explainer SHAP (see Figure 2c for the real image and Figure 2i for the fake image).

SHAP explanations are more focused on specific parts of images compared to XGBoost. However, it is still not easy to gain insights about which areas of an image are more important as pixels all over the image participate in the explanations of SHAP and XGBoost. For example, both XGBoost and SHAP distinguish some edge and middle pixels as key pixels (the bright pixels are more important) but it is not clear why these are important pixels.

*CXps enumeration approach.* We recall that our goal is to investigate whether there is a connection between the important pixels that SHAP/XGBoost finds and CXps for a given image. The most surprising result is that, indeed, a connection exists and, for example, it reveals that the edge pixels of an image, highlighted by both SHAP and XGBoost as important pixels, are, in fact, CXps of small cardinalities. Given all CXps of size $k$, we plot a heatmap of occurrences of each pixel in these CXps of size $k$. Let us focus on the first row with the real 6. Consider the heatmap CXp$^1$ at Figure 2d that shows all CXps of size one for the real 6. It shows that most of important pixels of XGBoost and SHAP are actually CXps of size one. This means that *it is sufficient to change a single pixel value to some other value to obtain a different prediction*. Note that these results reveal an interesting observation. DCGAN generates images with a few gray edges pixels (see Figure 4 in Appendix. Indeed, some of them have several edge pixels in gray.) This 'defect' does not happen often for real MNIST images. Therefore, the classifier 'hooks' on this issue to classify an image as fake. Now, consider the heatmap CXp$^2$ at Figure 2e of CXps of size two. It overlaps a lot with SHAP important pixels in the middle of the image explaining *why* these are important. Only a *pair* of these pixels can be changed to get a different prediction.

*A correlation between CXps and SHAP's important features.* To qualitatively measure our observations on correlation between key features of CXps and SHAP, we conducted the same experiment as

| | Dataset | | | | | |
|---|---|---|---|---|---|---|
| | **Adult** | **Lending** | **Recidivism** | **Compas** | **German** | **Spambase** |
| **# of instances** | 5579.0 | 4414.0 | 3696.0 | 778.0 | 1000.0 | 2344.0 |
| **total time (sec.)** | 7666.9 | 443.8 | 3688.0 | 78.4 | 16 943.2 | 6859.2 |
| **minimal time (sec.)** | 0.1 | 0.0 | 0.1 | 0.0 | 0.2 | 0.1 |
| **average time (sec.)** | 1.4 | 0.1 | 1.0 | 0.1 | 16.9 | 2.9 |
| **maximal time (sec.)** | 13.1 | 0.8 | 8.9 | 0.5 | 193.0 | 23.1 |
| **total oracle calls** | 492 990.0 | 69 653.0 | 581 716.0 | 21 227.0 | 748 164.0 | 176 354.0 |
| **minimal oracle calls** | 14.0 | 11.0 | 17.0 | 13.0 | 23.0 | 12.0 |
| **average oracle calls** | 88.4 | 15.8 | 157.4 | 27.3 | 748.2 | 75.2 |
| **maximal oracle calls** | 581.0 | 73.0 | 1426.0 | 134.0 | 7829.0 | 353.0 |
| **total # of AXps** | 52 137.0 | 8105.0 | 60 688.0 | 1931.0 | 59 222.0 | 18 876.0 |
| **average # of AXps** | 9.4 | 1.8 | 16.4 | 2.5 | 59.2 | 8.1 |
| **average AXp size** | 5.3 | 1.9 | 6.4 | 3.8 | 7.5 | 4.6 |
| **total # of CXps** | 66 219.0 | 8663.0 | 77 784.0 | 3558.0 | 66 781.0 | 24 774.0 |
| **average # of CXps** | 11.9 | 2.0 | 21.1 | 4.6 | 66.8 | 10.6 |
| **average CXp size** | 2.4 | 1.4 | 2.6 | 1.5 | 3.6 | 2.3 |

Table 1: Results of the computational experiment on enumeration of AXps and CXps.

above on 100 random images and measured the correlation between CXps and SHAP features. First, we compute a set $T$ of pixels that is the union of the first (top) 100 smallest size CXps. On average, we have 60 pixels in $T$. Note that the average 60 pixels represent a small fraction (7%) of the total number of pixels. Then we find a set $S$ of $|T|$ SHAP pixels with highest absolute weights. Finally, we compute $corr = |S \cap T|/|S|$ as the correlation measure. Note that $corr = 0.4$ on average, i.e. our method hits 40% of best SHAP features. As the chances of two tools independently hitting the same pixel (out of 784) are quite low, the fact that 40% of $|S|$ are picked indicates a significant correlation.

**Enumeration of CXps and AXps.** Here, we aim at testing the *scalability* of explanation enumeration and consider the six well-known and publicly available datasets. Three of them were previously studied in Ribeiro et al. (2018) in the context of heuristic explanation approaches, namely, Anchor Ribeiro et al. (2018) and LIME Ribeiro et al. (2016), including *Adult*, *Lending*, and *Recidivism*. Appendix C.2 provides a detailed explanation of datasets and our implementation. A prototype implementing is an adaptation of Liffiton & Sakallah (2008) abductive or (2) all contrastive explanations was created. In the experiment, the prototype implementation is instructed to enumerate all abductive explanations. The prototype is able to deal with tree ensemble models trained with XGBoost Chen & Guestrin (2016). Given a dataset, we trained an XGBoost model containing 50 trees per class, each tree having depth 3. (Further increasing the number of trees per class and also increasing the maximum depth of a tree did not result in a significant increase of the models' accuracy on the training and test sets for the considered datasets.) All abductive explanations for every instance of each of the six datasets were exhaustively enumerated using the duality-based approach (Algorithm 2 in Appendix B). This resulted in the computation of all contrastive explanations as well).

*Evaluation results.* Table 1 shows the results. There are several points to make. First, although it seems computationally expensive to enumerate all explanations for a data instance, it can still be achieved effectively for the medium-sized models trained for all the considered datasets. This may on average require from a few dozen to several hundred of oracle calls per data instance (in some cases, the number of calls gets up to a few thousand). Also observe that enumerating all explanations for an instance takes from a fraction of a second to a couple of seconds on average. These results demonstrate that our approach is practical.

Second, the total number of AXps is typically lower than the total number of their contrastive counterparts. The same holds for the average numbers of abductive and contrastive explanations per data instance. Third and finally, AXps for the studied datasets tend to be larger than contrastive explanations. The latter observations imply that contrastive explanations may be preferred from a user's perspective, as the smaller the explanation is the easier it is to interpret for a human decision maker. (Furthermore, although it is not shown in Table 1, we noticed that in many cases contrastive explanations tend to be of size 1, which makes them ideal to reason about the behaviour of an ML model.) On the other hand, exhaustive enumeration of contrastive explanations can be more time consuming because of their large number.

**Summary of results.** We show that CXps enumeration gives us an insightful understanding of a classifier's behaviour. First, even in cases when we cannot enumerate all of CXps to compute AXps by duality, we can still draw some conclusions, e.g. CXps of size one are exactly features that occur

in all AXps. Next, we clearly demonstrate the feasibility of the duality-based exhaustive enumeration of both AXps and CXps for a given data instance using a more powerful algorithm that performs enumeration of AXps and CXps.

## 5 CONCLUSIONS

This paper studies local model-based abductive and contrastive explanations. Abductive explanations answer 'Why?' questions, whereas contrastive explanations answer 'Why Not?' questions. Moreover, the paper relates explanations with the analysis of inconsistent theories, and shows that abductive explanations correspond to minimal unsatisfiable subsets, whereas contrastive explanations can be related with minimal correction subsets. As a consequence of this result, the paper exploits a well-known minimal hitting set relationship between MUSes and MCSes Reiter (1987); Birnbaum & Lozinskii (2003) to reveal the same relationship between abductive and contrastive explanations. In addition, the paper exploits known results on the analysis of inconsistent theories, to devise algorithms for extracting and enumerating abductive and contrastive explanations.

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

# Appendices

## A  TAXONOMY

The taxonomy of explanations used in the paper is summarized in Table 2.

Table 2: Taxonomy of ML model explanations used in the paper.

| | | **Instance-** | |
| | | *dependent* | *independent* |
|---|---|---|---|
| **ML model-** | agnostic | Heuristic *local* explanation for $\pi$. Examples: SHAP, LIME, Anchor, etc. | Heuristic *global* explanation for $\pi$. Examples: SHAP, LIME (e.g. submodular pick) |
| | based | Rigorous *local* explanation for $\pi$. Examples: | Rigorous *global* explanation for $\pi$. Examples: absolute/global AXps |

Within the "based / dependent" cell:

| 'Why $\pi$?' | 'Why not $\neg\pi$ ?' |
|---|---|
| PI- (abductive) explanations (AXps) | contrastive (CXps) (our work) |

## B  EXTRACTING & ENUMERATING EXPLANATIONS

The results of Section 3.3 enable exploiting past work on extracting and enumerating MCSes and MUSes to the setting of contrastive and abductive explanations, respectively. Perhaps surprisingly, there is a stark difference between algorithms for extraction and enumeration of contrastive explanations and abductive explanations. Due to the association with MCSes, one contrastive explanation can be computed with a logarithmic number of calls to a decision procedure Liffiton & Sakallah (2008). Moreover, there exist algorithms for the direct enumeration of contrastive explanations Liffiton & Sakallah (2008). In contrast, abductive explanations are associated with MUSes. As a result, any known algorithm for extraction of one abductive explanation requires at best a linear number of calls to a decision procedure Junker (2004), in the worst-case. Moreover, there is no known algorithm for the direct enumeration of abductive explanations, and so enumeration can be achieved only through the enumeration of contrastive explanations Liffiton & Sakallah (2008); Liffiton et al. (2016); Felfernig et al. (2012).

We adapt state-of-the-art algorithms for the enumeration MUSes and MCSes to find all the abductive and contrastive explanations. Note that as in the case of enumeration of MCSes and MUSes, the enumeration of CXps is comparatively easier than the enumeration of AXps. Algorithm 1 shows our adaptation of MCS enumeration algorithm to the enumeration of CXps Liffiton & Sakallah (2008). Other alternatives Grégoire et al. (2018) could be considered instead. Algorithm 1 finds a CXp, blocks it and finds the next one until no more exists. To extract a single CXp, we can use standard algorithm, e.g. Bailey & Stuckey (2005). In principle, enumeration of AXps can be achieved by computing all CXps and then computing all the minimal hitting sets of all CXps, as proposed in the propositional setting Liffiton & Sakallah (2008). However, there are more efficient alternatives that we can adapt here Bailey & Stuckey (2005); Liffiton et al. (2016); Narodytska et al. (2018); Bendík et al. (2018), Algorithm 2 adapts Liffiton et al. (2016) to the case of computing both AXps and CXps. The algorithm simultaneously searches for AXps and CXps and is based on the hitting set duality.

---

**Algorithm 1** Enumeration of CXps

**Function** CXPENUM ($\mathcal{M}_\pi$,$\mathcal{C}$, $\pi$)
    **Input:** $\mathcal{M}_\pi$: ML model, $\mathcal{C}$: Input cube, $\pi$: Prediction
    **Variables:** $\mathcal{N}$ and $\mathcal{P}$ defined on the variables of $\mathcal{C}$

1    $\mathcal{I} \leftarrow \emptyset$ ;                       // Block CXps
2    **while** true **do**
3        $\mu \leftarrow$ ExtractCXp($\mathcal{M}_\pi$,$\mathcal{C}$,$\pi$,$\mathcal{I}$)
4        **if** $\mu = \emptyset$ **then break**;
5        ReportCXp($\mu$)
6        $\mathcal{I} \leftarrow \mathcal{I} \cup$ NegateLiteralsOf($\mu$)

---

---

**Algorithm 2** Enumeration of AXps (and CXps)

---

**Function** XPENUM ($\mathcal{M}_\pi, \mathcal{C}, \pi$)
    **Input:** $\mathcal{M}_\pi$: ML model, $\mathcal{C}$: Input cube, $\pi$: Prediction
    **Variables:** $\mathcal{N}$ and $\mathcal{P}$ defined on the variables of $\mathcal{C}$

1    $\mathcal{K} = (\mathcal{N}, \mathcal{P}) \leftarrow (\emptyset, \emptyset)$ ;                                        // Block AXps & CXps
2    **while** true **do**
3        $(st_\lambda, \lambda) \leftarrow$ FindMHS$(\mathcal{P}, \mathcal{N})$ ;                           // MHS of $\mathcal{P}$ st $\mathcal{N}$
4        **if** $\neg st_\lambda$ **then break**;
5        $(st_\rho, \rho) \leftarrow$ SAT$(\lambda \wedge \neg \mathcal{M}_\pi)$
6        **if** $\neg st_\rho$ **then**                                   // entailment holds
7            ReportAXp$(\mu)$
8            $\mathcal{N} \leftarrow \mathcal{N} \cup$ NegateLiteralsOf$(\mu)$
9        **else**
10       $\mu \leftarrow$ ExtractCXp$(\mathcal{M}_\pi, \rho, \pi)$
11       ReportCXp$(\rho)$
12       $\mathcal{P} \leftarrow \mathcal{P} \cup$ UseLiteralsOf$(\rho)$

---

## C   ADDITIONAL EXPERIMENTAL RESULTS

### C.1   ENUMERATION OF CXPS

**Setup.**   To perform enumeration of contrastive explanations in our first experiment, we use a constraint programming solver, ORtools Perron & Furnon [15]. To encode the enumeration problem with ORtools we converted scores of XGBoost models into integers keeping 5 digits precision. We enumerate contrastive explanations in the increasing order by their cardinality. This can be done by a simple modification of Algorithm 1 forcing it to return CXps in this order. So, we first obtain all minimum size contrastive explanations, and so on.

**Second experiment.**   Consider our second the "3 vs 5 digits" experiment. We use a dataset that contains digits "3" (class 0) and "5" (class 1) from the standard MNIST (see Figure 3a and Figure 3g for representative samples). XGboost model has 50 trees of depth 3 with accuracy 0.98 (0.97) on train/test sets. We quantized images so that each pixel takes a value between 0 and 15. As before, each pixel corresponds to a feature. So, we have 784 features in our XGBoost model.

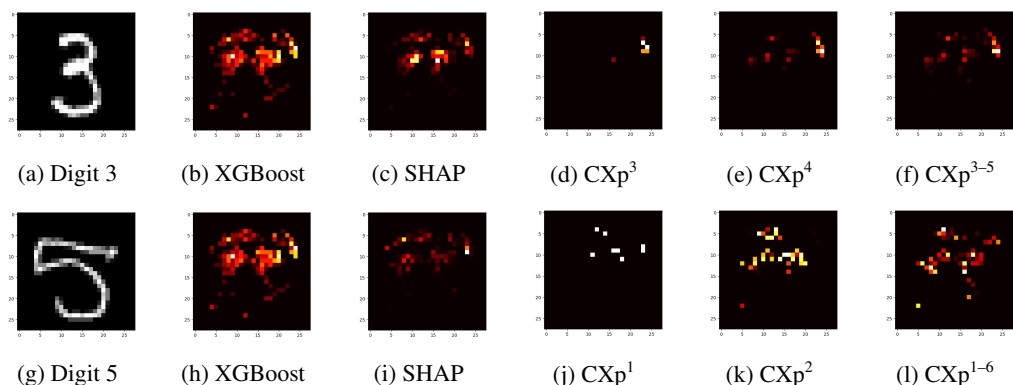

(a) Digit 3    (b) XGBoost    (c) SHAP    (d) CXp$^3$    (e) CXp$^4$    (f) CXp$^{3-5}$

(g) Digit 5    (h) XGBoost    (i) SHAP    (j) CXp$^1$    (k) CXp$^2$    (l) CXp$^{1-6}$

Figure 3: Results of the 3 vs 5 digits experiments. The first row shows results for the image 3. The second row shows results for the image 5. The first column shows examples of inputs; the second column shows heatmaps of XGBoost's global important features; the third column shows heatmaps of SHAP's important features. Last three columns show heatmaps of CXp of different cardinality.

**Global and local explainers.**   We start by discussing our results on few random samples (Figure 3a and Figure 3g). First, we obtain the important features from XGBoost. As these features are *global* for the model so they are the same for all inputs (Figure 3b and Figure 3h are identical for 3 and 5

---

[15]The prototype and the experimental setup are available at https://github.com/xdual/xdual.

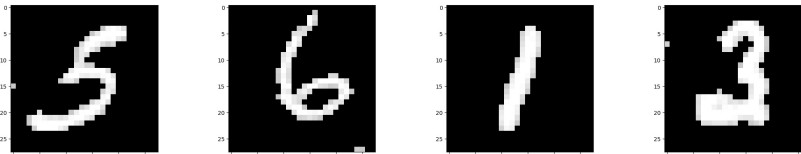

Figure 4: Additional fake images. We reduced values of zero-valued pixels to highlight gray pixels on the edges for some fake images.

images). Figure 2b shows that these important features. The important pixels highlight that the top parts of images are important, which is a plausible high-level explanation of the classifier behavior. Digits 3 and 5 are mostly differ in the top part of the image. However, some pixels are way more important than other and it is hard to understand why.

Next, we compute an image-specific explanation using the standard explainer SHAP ( see Figure 3c for the digit 3 and Figure 3c for the digit 5). While SHAP explanations mimic XGBoost important features, they do provide additional insights for the user. Note that both XGBoots and SHAP mark a "belt" of pixels in the upper middle part that as important (bright pixels is the most important pixels).

**CXps enumeration approach.** We run our enumeration algorithm to produce CXps of increasing cardinality. For each image, we enumerate first 2000 CXps. Given all CXps of size $k$, we plot a heatmap of occurrences of each pixel in these CXps of size $k$. Let us focus on the second row with the digit 5. For example, $CXp^2$ (Figure 3k) shows the heatmap of CXps of size two for the digit 5. As we mentioned above, both XGBoost and SHAP hint that the 'belt' of important pixels in the middle. Again, our method can explain *why* this is the case. Consider the heatmap $CXp^1$ at Figure 3j. This picture shows all CXps of size one for the digit 5. It reveals that most of important pixels of XGBoost and SHAP are actually CXps of size one. We reiterate that it is sufficient to change a *single* pixel value to some other value to obtain a different prediction. Now, consider the heatmap $CXp^{1-6}$ at Figure 3l. This figure shows 2000 CXps (from size 1 to size 6). It overlaps a lot with SHAP important pixels in the middle of the image. So, these pixels occur in many small size CXps and changing their values leads to misclassification.

**Correlation between CXps and SHAP features.** To qualitatively measure our observations on correlation between key features of CXps and SHAP, we conducted the same experiment as above on 100 random images and measured the correlation between CXps and SHAP features. First, we compute a set $T$ of pixels that is the union of the first (top) 100 smallest size CXps. On average, we have 38 pixels in $T$. Note that the average 38 pixels represent a small fraction (5%) of the total number of pixels. Then we find a set $S$ of $|S|$ SHAP pixels with highest absolute weights. Finally, we compute $corr = |S \cap T|/|S|$ as the correlation measure. Note that $corr = 0.6$ on average, i.e. our method hits 60% of best SHAP features. As the chances of two tools independently hitting the same pixel (out of 784) are quite low, the fact that 60% of $|T|$ are picked indicates a significant correlation.

## C.2 ENUMERATION OF CXPS AND AXPS

**Datasets.** The results are obtained on the six well-known and publicly available datasets. Three of them were previously studied in Ribeiro et al. (2018) in the context of heuristic explanation approaches, namely, Anchor Ribeiro et al. (2018) and LIME Ribeiro et al. (2016), including *Adult*, *Lending*, and *Recidivism*. These datasets were processed the same way as in Ribeiro et al. (2018). The *Adult* dataset Kohavi (1996) is originally taken from the Census bureau and targets predicting whether or not a given adult person earns more than $50K a year depending on various attributes, e.g. education, hours of work, etc. The *Lending* dataset aims at predicting whether or not a loan on the Lending Club website will turn out bad. The *Recidivism* dataset was used to predict recidivism for individuals released from North Carolina prisons in 1978 and 1980 Schmidt & Witte (1988). Two more datasets were additionally considered including *Compas* and *German* that were previously studied in the context of the FairML and Algorithmic Fairness projects FairML; Friedler et al. (2015); Feldman et al. (2015); Friedler et al. (2019), an area in which the need for explanations is doubtless. *Compas* is a popular dataset, known Angwin et al. (2016) for exhibiting racial bias of the COMPAS algorithm used for scoring criminal defendant's likelihood of reoffending. The latter dataset is a German credit data (e.g. see Feldman et al. (2015); Friedler et al. (2019)), which given a list of people's attributes classifies them as good or bad credit risks. Finally, we consider the *Spambase*

dataset from the UCI repository Dua & Graff (2017). The main goal is to classify an email as spam or non-spam based on the words that occur in this email. Due to scalability constraints, we preprocessed the dataset to keep ten words per email that were identified as the most influential words by a random forest classifier.

**Implementation and Setup.** A prototype implementing Algorithm 2 targeting the enumeration of either (1) all abductive or (2) all contrastive explanations was created. In the experiment, the prototype implementation is instructed to enumerate all abductive explanations. (Note that, as was also mentioned before, no matter what kind of explanations Algorithm 2 aims for, all the dual explanations are to be computed as a side effect of the hitting set duality.) The prototype is able to deal with tree ensemble models trained with XGBoost Chen & Guestrin (2016). For that purpose, a simple encoding of tree ensembles into satisfiability modulo theories (SMT) was developed. Concretely, the target formulas are in the theory of linear arithmetic over reals (RIA formulas). (Note that encodings of a decision tree into logic are known Bonfietti et al. (2015); Lombardi et al. (2017); Verwer et al. (2017). The final score summations used in tree ensembles can be encoded into RIA formulas.)

Due to the twofold nature of Algorithm 2, it has to deal with (1) implicit hitting set enumeration and (2) entailment queries with SMT. The former part is implemented using the well-known MILP solver CPLEX IBM ILOG. SMT solvers are accessed through the PySMT framework Gario & Micheli (2015), which provides a unified interface to a variety of state-of-the-art SMT solvers. In the experiments, we use Z3 de Moura & Bjørner (2008) as one of the best performing SMT solvers. The conducted experiment was performed in Debian Linux on an Intel Xeon E5-2630 2.60GHz processor with 64GByte of memory.

