# OpenReview forum: "On Relating "Why?" and "Why Not?" Explanations"
_ICLR.cc/2021/Conference — Reject_

### Official Review · AnonReviewer3 · 2020-10-28

**Rating:** 5
**Confidence:** 2

**Review:**

I thank the authors for their submission. I believe the investigated content is relevant and timely and would perhaps benefit from the discussion in a community such as the one of ICLR. Please find my comments below, as potential points of discussion.

High-level comments:
* generally, the paper does a good review of existing literature and aims to relate two important subfields (abductive vs contrastive explanations) using rules of logic, particularly using the minimal hitting set relationship
* beyond showing that such a duality holds between abductors and contrastive explanations, I believe the experimental section should further explore comparisons with related work such as Dhurandhar et al. (as cited in earlier sections) and Rebeiro et al.
* furthermore, the idea of using FOL for generative contrastive explanations (also sometimes called counterfactual explanations) has been explored before (e.g., [Karimi et al.]);
* on the presentation of material, there seemed to be an underwritten requirement to be familiar and have a background in logic and verification, which makes me wonder whether ICLR is the right venue here (I also apologize for not being to provide much feedback on the technical front)

Minor comments and nits:
- the footnotes seemed to contain important details, but the number of footnotes seemed overwhelming and hurt the flow of reading
- [footnote 9] seems incorrect; e.g., non-linearities in MLPs or RBF kernels in SVMs cannot be encoded as first-order logic
- perhaps figure 2 can be redone to more visually demonstrate the benefit of using the proposed method; if the provided explanations aren’t visually appealing (relatively), then perhaps consider a non-image-based dataset?

[Ribeiro et al.] https://homes.cs.washington.edu/~marcotcr/aaai18.pdf
[Karimi et al.] https://arxiv.org/abs/1905.11190

---

> ### Author Response · Authors · 2020-11-14
> **On experiments, comparisons and FOL**
>
> **Response:**
> Thank you for the thoughtful review.
> The main purpose of the experiments was to demonstrate that enumeration of explanations can scale to realistic ML models. In addition, we analyzed SHAP aiming to illustrate possible uses of our work. We did try Anchor, but the tool was too slow for the examples we considered. As a result, we focused on analyzing SHAP. The same general comment applies to the work of Dhurandhar et al., i.e. the conclusions would be aligned with what we observed for SHAP.
>
> Thank you for the pointer to the work of [Karimi et al.]. This work will be cited, as it also exploits FOL to represent ML models, and it tackles contrastive explanations. However, our work not only offers a simpler and more efficient approach for computing contrastive explanations, it also proves the hitting set relationship between the two kinds of explanations. Our claim of being more efficient is
> justified by the results presented in [Karimi et al.], which scales to a few ReLU units, by the results presented in our paper, but also by the results presented in the papers our work builds upon.
>
> We feel ICLR is a suitable venue for our work. There are several papers on using formal methods in ML that have been published at ICLR in recent years.
>
> Finally, regarding footnote 9, we note that there is nothing wrong with it. It is well-known that one can encode non-linear arithmetic in FOL. This is actually available in a wide range of SMT solvers. We can add a comment about this to the paper.
>
> **What to change in the paper:**
> - We will include the missing reference to [Karimi et al.]. Thank you again for spotting this.
> - We will improve the presentation of results, as suggested by this and another reviewer.

---

### Official Review · AnonReviewer1 · 2020-10-30
**develops contrastive explanations using first order logic**

**Rating:** 6
**Confidence:** 2

**Review:**

Overview
This paper develops, what it calls, contrastive explanations using first order logic. Contrastive explanations answer the why not question, while standard explanations (abductive in this paper's language), answer why. This paper formulates the problem as a first order logic problem and then leverages what seems like classic algorithms from that domain to learn these explanations. First, I want to point out that I have no expertise in first order logic, so this review is perhaps an educated guess as to the quality of this work.

Clarity/Writing
The writing is relatively clear and straightforward. One issue is with the citations: there is a missing bracket with the citations, which then makes them part of the sentence and ultimately distracting.

Quality/Significance.
Contrastive explanations could be useful for a variety of tasks and importantly provide actionable insights about how to change a sample to satisfy positive rating. In that regard, the goal of this work is a worthy one. I cannot judge the quality of this work since I donot have any expertise in formal logic.

Questions
What is the difference between the contrastive explanations as defined here and what watcher et. al. and the literature calls counterfactual explanations?

I am somewhat surprised at the computational performance from the results presented here; these seem much faster than I would have expected. It seems like the task of identifying a contrastive explanation, for the discretized data set, should be NP-Hard. Is this the case?

In looking at figure 2, I can't really see the differences in attribution for CXp for the real data vs the fake one. How does the CXp help distinguish real from fake data? In the summary of results, the authors conclude that the CXps enumeration provide insight into the behavior of the classifier. However, the behavior is not discussed nor is any insight provided. What are the authors referring to?

---

> ### Author Response · Authors · 2020-11-14
> **Clarifications on definitions and NP-hardness**
>
> **Response:**
> Thank you for the positive comments. We opted to refer to contrastive explanations following the work of Miller.
>
> Regarding the performance, the fact that we are solving NP-hard problems does not necessarily mean those problems cannot be solved efficiently in practice. The last two decades of research in automated reasoning demonstrated that some NP-hard/NP-complete problems can be efficiently solved in practice, at least in the vast majority of problems one encounters in practice.
>
> We will improve the presentation of results.
>
> **What to change in the paper:**
> We will improve the presentation of results, addressing the reviewer's concerns.

---

### Official Review · AnonReviewer2 · 2020-10-30
**Compelling idea but limited setting and experiments**

**Rating:** 5
**Confidence:** 5

**Review:**

The authors propose to extract two types of explanations: abductive and contrastive explanations to address a gap in the literature of explainable AI. Indeed, that's a great point and often explainable models address the "why" and rarely the "why not" that can help identify the features guiding the change in the class.
The ideas presented are compelling and it is good to see that we can re-use state of the art AI first order logic (FOL) statements in the field of explainable models. The references are excellent.
However, the paper suffers from several drawbacks:
1- The setting is limited to ML models that are expressed as a set of FOL sentences
2- Discretization of numerical features is required. We know static discretization can be problematic (large versus small interval); no discussion is in the paper to how to address this point
3- More details should be presented about SHAP since this is the main method the authors compare to;
4- Figure 1 part c) is not explained. There seems to be missing parentheses in the FOL statement;
5- The experimental section is rather weak.  The example on the real-vs-fake digit is not clear. Pointing out the brighter pixels as those responsible of the classification is not convincing to me. Comparison to Shap (using correlation) is not discussed.
The second experiment. provides some statistics on the time, number of abductive and contrastive explanations. Perhaps it would have been good to provide other examples of the importance of extracting both explanations and how pertinent they are. Overall, there is a need for some baseline to validate the explanations (both types);
6- The methodology should be made clearer, notations introduced or re-introduced, many readers might not be familar with some notations like entailments;

Overall, I feel there are good ideas in there, the authors should consider enlarging the spectrum of the applicability of their approach, may be rework the definitions and methodology section and design more solid experiments.

Minor comments:
seeFigure

---

> ### Author Response · Authors · 2020-11-14
> **On limited setting and experiments.**
>
> **Response:**
> Thank you for the detailed review.
> Replies to the identified drawbacks:
> 1. We ask the reviewer to clarify why FOL might be perceived as a limitation. As commented in the response to reviewer 4, FOL tools are fairly flexible in what they allow modeling; this includes non-linear arithmetic. For example, a wide class of neural networks with piecewise linear activation functions, boosted decision trees, etc., can be easily expressed using FOL. As we mentioned above, we can express models with nonlinear activation functions as well.
>
> 2. Technically, we do not need to discretize features. Note that the proposed approach is perfectly general and *constraint-agnostic*, e.g. it allows us to reason with real or integer-valued features (at the potential cost of efficiency and/or decidability). The computation of prime implicants when reasoning about real-valued domains has been studied before, e.g. [*Isil Dillig, Thomas Dillig, Kenneth L. McMillan, Alex Aiken: Minimum Satisfying Assignments for SMT. CAV 2012: 394-409*](https://doi.org/10.1007/978-3-642-31424-7_30). However, we should note that this was not the goal of the paper. The goal is to develop formal approaches for computing and enumerating contrastive explanations and, more importantly, to demonstrate the hitting set duality between the two types of explanations.
> 3. Additional details regarding SHAP will be added to the paper.
> 4. We will complete the analysis of Figure 1. Thanks for noticing this.
> 5. The purpose of the experiments was twofold: (1) to demonstrate that enumeration of explanations can scale to realistic ML models and (2) to get a better perception of what SHAP's explanations correspond to using the proposed apparatus of rigorous contrastive explanations - this is to demonstrate possible uses of our work. We note that the purpose of this analysis was not to *"compare to SHAP"*. We will clarify the digit example. We re-emphasize that the objective was to show scalability and applicability, which we did.
> 6. We will improve readability for the final version.
>
> **What to change in the paper:**
> - We will add detail on SHAP.
> - We will complete the description of Figure 1.
> - We will improve the presentation of the experiments.

---

### Official Review · AnonReviewer4 · 2020-10-30
**A formal framework to understand contrastive and abductive explanations**

**Rating:** 8
**Confidence:** 3

**Review:**

Summary:
-------------
This work presents first of a kind logic-based framework that relates contrastive (minimally absent) and abductive (minimally present) explanations and shows that abductive explanations are essentially minimal hitting sets of contrastive explanations.

+ve:
-----
A much needed formal framework and proofs that relate the different types of explanations. With many different types of explanations proposed in the XAI literature, this line of work improves are understanding of the overall space & taxonomy of explanations. Also the relationship between types of explanations may helps us enumerate/compute other types of explanations based on the computation of one type.

Suggestions:
-------------------
It would be good to comment a bit on the overall steps needed to convert any ML problem into the proposed framework - costs of binarizing feature values, etc. before applying the proposed ideas for local or global explanations.

---

> ### Author Response · Authors · 2020-11-13
> **Reviewer's suggestions will be addressed**
>
> **Response:**
> Thank you for the positive review.
> We will address the suggestions made.
>
> **What to change in the paper:**
> We will add a paragraph explaining how one can encode different ML
> models into a logic-based representation, amenable to formal
> reasoning. We will also briefly comment on the differences between
> tackling local and global explanations.

---

### Decision · Program_Chairs · 2021-01-07
**Final Decision**

**Decision:**

Reject

**Comment:**

The authors consider local 'why' or 'abductive' explanations for a model and a given class, which identify a minimal subset of features such that they're sufficient to imply that the model predicts the class; and 'why not' or 'contrastive' explanations, which identify a minimal subset s.t. they're sufficient to imply that the model predicts a different class. The two types of explanation are related using earlier work on minimal hitting sets going back to Reiter (1987).

Reviewers were divided in their opinions. R4 was very positive but with little detail and only medium confidence, then did not participate in discussion. R2 was the only reviewer with high confidence, leaning against acceptance. The paper relies on FOL which was hard for reviewers to grapple with, and may make it challenging for readers. The presentation could be improved by clearly linking to existing work and demonstrating why the new approach is important.